# Genomic Assessment of Cancer Susceptibility in the Threatened Catalina Island Fox (*Urocyon littoralis catalinae*)

**DOI:** 10.3390/genes13081496

**Published:** 2022-08-22

**Authors:** Sarah A. Hendricks, Julie L. King, Calvin L. Duncan, Winston Vickers, Paul A. Hohenlohe, Brian W. Davis

**Affiliations:** 1Institute for Interdisciplinary Data Sciences, University of Idaho, Moscow, ID 83844, USA; 2Catalina Island Conservancy, P.O. Box 2739, Avalon, CA 90704, USA; 3Institute for Wildlife Studies, Arcata, CA 95521, USA; 4Karen C. Drayer Wildlife Health Center, School of Veterinary Medicine, University of California, Davis, CA 95616, USA; 5Department of Biological Sciences, University of Idaho, Moscow, ID 83844, USA; 6Department of Veterinary Integrative Biosciences, College of Veterinary Medicine and Biomedical Science, Texas A&M University, College Station, TX 77840, USA; 7Department of Small Animal Clinical Sciences, College of Veterinary Medicine and Biomedical Science, Texas A&M University, College Station, TX 77840, USA

**Keywords:** cancer, fox, conservation, genomics, evolutionary rescue, population bottleneck, genetic drift

## Abstract

Small effective population sizes raise the probability of extinction by increasing the frequency of potentially deleterious alleles and reducing fitness. However, the extent to which cancers play a role in the fitness reduction of genetically depauperate wildlife populations is unknown. Santa Catalina island foxes (*Urocyon littoralis catalinae*) sampled in 2007–2008 have a high prevalence of ceruminous gland tumors, which was not detected in the population prior to a recent bottleneck caused by a canine distemper epidemic. The disease appears to be associated with inflammation from chronic ear mite (*Otodectes*) infections and secondary elevated levels of *Staphyloccus pseudointermedius* bacterial infections. However, no other environmental factors to date have been found to be associated with elevated cancer risk in this population. Here, we used whole genome sequencing of the case and control individuals from two islands to identify candidate loci associated with cancer based on genetic divergence, nucleotide diversity, allele frequency spectrum, and runs of homozygosity. We identified several candidate loci based on genomic signatures and putative gene functions, suggesting that cancer susceptibility in this population may be polygenic. Due to the efforts of a recovery program and weak fitness effects of late-onset disease, the population size has increased, which may allow selection to be more effective in removing these presumably slightly deleterious alleles. Long-term monitoring of the disease alleles, as well as overall genetic diversity, will provide crucial information for the long-term persistence of this threatened population.

## 1. Introduction

As threats to wild populations increase due to habitat destruction and climate factors, understanding the genetic consequences affecting small, isolated populations can inform conservation practices to mitigate the continued loss of genetic diversity. When the effective population size is small, both inbreeding and stochastic processes such as genetic drift can increase the frequency of deleterious alleles. Increased genetic load is particularly difficult to eliminate from small populations because drift hinders the power of selection to remove weakly and moderately deleterious alleles, as has been observed in island foxes [1,2]. These deleterious alleles become fixed with time and lead to mutational meltdown as fitness is reduced and the probability of extinction is increased [3,4,5,6]. Alternatively, bottlenecks resulting in reduced effective population size can increase the frequency of deleterious mutations transiently, though the mutational load may persist as heterozygotes [7,8]. These deleterious alleles may still reduce fitness in many ways, including increased disease susceptibility [9]. However, the role of deleterious variants that specifically influence cancer susceptibility is poorly explored [10].

Studies of cancer in wildlife species present unique challenges, but research is beginning to uncover the impact of cancer on wildlife populations [11,12,13]. Population genomic tools are increasingly being used to investigate wildlife cancers, addressing issues such as the genetic variation for susceptibility within populations, comparative genomics of tumor suppressor genes, and evolutionary response to cancers, as reviewed in [14]. Cancer can affect wild populations by reducing reproductive success, altering population dynamics, and directly or indirectly leading to population declines [11,13]. As well, pathological inflammation from genetic or environmental causes manifesting in pre- and peri-reproductive individuals can negatively affect reproductive fitness [15]. In addition to presenting a major conservation concern, naturally occurring cancers in wildlife species may provide new biological models for understanding the often complex genetic and environmental etiologies of cancer, with the potential for biomedical benefits to domestic animals and humans [14]. Known causes of cancer in wildlife include environmental carcinogens (e.g., polycyclic aromatic hydrocarbons), viruses and other pathogens, direct transmission of tumor cells, and hereditable factors [11,14].

Few genetic studies of wild populations have attempted to identify polymorphisms putatively associated with cancer susceptibility [16,17,18,19]. For example, a study in California sea lions demonstrated that urogenital carcinoma was significantly associated with homozygosity of a microsatellite locus within an intron of the heparanase 2 gene *HPSE2* [16], and facial tumor disease regression is associated with *RAS11A* activation in Tasmanian devils [18], both of which are implicated in several human carcinomas. Among mammalian species, multiple genes associated with cancer show evidence of germline positive selection [20,21,22,23]. However, in general, studying the genetics of cancer in wild systems is especially challenging given difficulties in sampling combined with the ethical, logistical, and legal limits on experimentation. 

The island fox (*Urocyon littoralis*) (Figure 1A) is a species related to the mainland gray fox (*Urocyon cinereoargenteus*) that was thought to have colonized the three northern Channel Islands near the coast of southern California about 10,000–16,000 years bp, with the first evidence of colonization of the southern three islands by mainland or island populations 2000–4000 years bp [24,25]. More recent evidence indicated more recent fox colonization of the northern islands ~9200–7100 years ago [26]. The number of foxes on these isolated islands varies from a few hundred to ~2000 individuals, depending on island size, and the density of individuals varies within each island [24,27,28]. Four of the island populations have experienced severe genetic bottlenecks in the late 1990s due to golden eagle predation on three northern islands (Santa Rosa, Santa Cruz, and San Miguel) and a canine distemper virus (CDV) epidemic on Santa Catalina Island (SCA) that caused an 85% population reduction, resulting in these four subspecies being federally listed as endangered in 2004 [29,30]. Subsequently, these populations rebounded under human management in the fastest recovery of any mammal under the Endangered Species Act to date [30,31]. High genetic variation in San Nicholas Island (SNI) foxes (*Urocyon littoralis dickey*) was subsequently discovered for the major histocompatibility complex (MHC) and was hypothesized to be maintained by balancing selection [32]. However, subsequent reanalysis of these data indicates that the two DRB alleles seen in SNI are the most common of the three alleles in SCA, indicating founder effects or drift could account for this difference [33]. As well, the heterozygosity for DQB was consistent with nonselective effects, reducing the likelihood of balancing selection having a considerable effect on the MHC [33]. Lastly, island fox populations do not appear to suffer from severe inbreeding depression in aggregate [1,2] and exhibit putatively adaptive differentiation among islands despite very low levels of overall genetic diversity [34].

SCA foxes have a high prevalence (~50 percent of individuals four years or older sampled in 2007–2008) of ceruminous gland carcinoma and adenoma (collective tumors of the ear canal) that appear to be associated with inflammation induced by chronic ear mite (Otodectes) infections [35]. Ceruminous gland tumors have not been documented on two geographically close islands (San Clemente Island; SCLE and San Nicolas Island; SNI) despite similar levels of chronic mite infection, and they are also not observed in the three island fox populations that have no ear mites (Santa Cruz, Santa Rosa, San Miguel) [35]. Further, environmental factors such as aerobic and anaerobic bacteria, yeast, fungi, viruses (herpesviruses and papillomaviruses), and toxic chemicals (polychlorinated biphenyls and organochlorines) are not associated with the presence of tumors [35]. However, secondary infection by *Staphylococcus pseudintermedius*, an opportunistic pathogen, has been shown to influence the sustained inflammation associated with tumor development [35]. Ear mite infections are often acquired soon after birth from maternal contact and persist indefinitely unless treated, and the inflammatory response associated with tumorgenesis is unique to SCA [36]. However, the onset of disease, though present in young foxes, is the highest burden in mature foxes. This indicates that the potential selection pressure may be moderate, as foxes often reproduce in their first year [37], though inflammatory conditions have been shown to affect reproductive fitness when manifesting in pre-reproductive individuals [15]. Although environmental factors cannot yet be ruled out and genetic determinism has yet to be demonstrated, this variation in tumor prevalence allows a unique opportunity to investigate the genetic influences on cancer susceptibility in a set of small, closely related yet isolated natural populations. 

Here, we test the hypothesis that the remarkably high incidence of ceruminous gland tumors in SCA island foxes is the result of genetic variants for cancer susceptibility that differ in frequency between cohorts using whole-genome sequencing of affected individuals (cases) and long-lived unaffected individuals (controls) from SCA, and control individuals from SCLE. We predict that the alleles that increase cancer susceptibility are deleterious and, therefore, relatively young and that they rose in frequency in the SCA population as they rebounded from the recent population bottleneck. Within the genomic regions of implicated loci, we expect reduced nucleotide diversity in cases, increased differentiation in allele frequencies between cases and controls, and reduced heterozygosity in cases. Along with these population genomic signatures, we use functional gene annotation from the domestic dog to identify candidate genes that contribute to cancer susceptibility in the SCA island fox.

## 2. Materials and Methods

### 2.1. Library Construction and Sequencing

We sequenced the genomes of 32 island foxes from Santa Catalina Island and San Clemente Island. The dataset included cases as individuals with carcinoma (*n* = 12), and control individuals from SCA that did not develop tumors (*n* = 11), see [35] for aging methods), and control individuals from SCLE (*n* = 9). Two of the SCLE individuals were previously published [1,2]. SCLE individuals provide an alternative control group to SCA as this fox population has not shown evidence of ceruminous gland tumors.

We obtained existing tissue and blood samples collected for a previous study [35]. DNA was extracted with a commercial kit using standard protocols (Qiagen DNA QiaAmp minikit, Hilden, Germany). Library preparation was completed at the Vincent J. Coates Genomic Sequencing Laboratory at the University of California, Berkeley, and libraries were run on 16 lanes with 150 bp paired-end sequencing on an Illumina HiSeq4000.

### 2.2. Computational Methods

Reads were trimmed using trimmomatic [38] and aligned to the domestic dog genome canFam3.1 [34] using SpeedSeq v0.12 [39]. Duplicate reads were removed using Picard 2.27 (broadinstitute.github.io/picard/, accessed on 1 November 2021). Aligned reads that were properly paired, mapped uniquely, and had high quality (Phred ≥ 30 for all bases) were used for variant calling using Genome Analysis Toolkit v4.2 [40] HaplotypeCaller (mapping quality ≥ 20), and the cohort was joint genotyped using GATK GenotypeGVCFs. 

### 2.3. Variant Filtration

Variant filtration was necessary to ensure sequencing data with high accuracy were used. Qualimap2 [41] was used to assess read quality. Only single nucleotide variants (SNV) on autosomes were used due to divergence between *Urocyon* and *Canis* genomes. SNV with a depth of coverage less than 60% or greater than 250% of the average for the individual were excluded since they may represent interspecies paralogy or copy number variation between domestic dog and island fox. SNV with minimum genotype quality of less than 20 and sites called in fewer than 95% of individuals or a minor allele frequency less than 0.01 were removed from further analysis. The number of segregating sites and mean coverage per individual were calculated using VCFtools [42]. We used KING v2.2 to calculate relatedness [43] and PRIMUS v1.9 to identify the maximum set of unrelated individuals from the filtered dataset [44,45]. Individuals were assessed for a kinship coefficient threshold greater than 0.0884 (related by 2nd degree or higher). Functional regions were annotated using SnpEff v5.1 [46] based on Ensembl release 106 for canFam3.1. We annotated variants within coding regions with respect to their effect on the amino acid sequence and polarized alleles as ancestral or derived using the domestic dog as an outgroup. 

### 2.4. Demographic Estimates

To assess genetic clustering, we applied two methods to the dataset of putatively neutral loci pruned for linkage disequilibrium (LD). Datasets were LD-pruned using PLINK v2.0 (-indep-pairwise 50 5 0.2) [47]. The dataset was pruned for strong LD (removing any SNV having a multiple r^2^ > 0.90 with all other SNV in a 50-SNV window) using PLINK, which reduced false autozygosity calls by removing redundant markers in SNV-dense regions and making SNV coverage more uniform. Coordinates of putatively neutral regions in the canine genome were obtained from Freedman et al. [48]. First, principal components analysis (PCA) was performed using PLINK v2.0. Second, we inferred a phylogenetic network with the neighbor-net method in SplitsTree4 v4.18.2 using uncorrected P distance [49].

Using the full, filtered dataset, we estimated historical demography using SMC++ v1.15.2 [50], which jointly estimates population histories and divergence times with unphased data. We assumed a mutation rate of 4.0 × 10^−9^/site/generation [51] and tested if the mutation rate affected divergence times by using a mutation rate of 1.0 × 10^−8^/site/generation (from [48]) and 2.0 × 10^−8^/site/generation [2]. Island foxes typically breed by the end of their first year [37]. Therefore, we used a generation time of 2 years to convert the coalescent scaling to calendar time.

### 2.5. Identification of Implicated Polymorphisms

We calculated several population genetic statistics to test for deviation from neutral expectation. Using Tajima’s D, we measured the departure from neutral evolution for each cohort with the hypothesis that a smaller Tajima’s *D* in cases would indicate an excess of rare variants compared to the controls, as has been performed in multiple heritable disease studies [52,53,54]. We also used nucleotide diversity per population (π) as well as Weir’s F_ST_ for each comparison pair (case/control, control/SCLE, case/SCLE). All were calculated using a sliding window approach in VCFtools. Z-score transformations were performed using custom R scripts using the SciPy library (https://github.com/scipy/scipy/, accessed on 1 November 2021). We calculated each statistic with a window size of 100 kb, step size of 10 kb, and at a finer scale assessment using 50 kb windows and 1 kb steps.

The cross-population composite likelihood ratio test XP-CLR was employed to compare allele frequency differentiation between comparison pairs to identify regions of elevated differentiation [55]. XP-CLR scores were estimated using the following parameters: -w1 0.0005 600 50000 –p1 0.95. A set of grid points as the putatively selected allele positions were positioned along each chromosome with a spacing of 50 kb. The sliding window around each grid point was set to 0.05 cM with a maximum number of SNV within each window set to 600. The correlation level (high LD) from which the SNV contributed to XP-CLR scores was down-weighted to 0.95. Third, XP-CLR scores were normalized with Z-scores using a custom python script using SciPy. The cross-population extended haplotype homozygosity XP-EHH test was also employed using the algorithm employed in [56] with default settings, and the unstandardized XP-EHH scores were transformed into a normal distribution using R version 4 [57].

The top 1% outlier windows for the difference in nucleotide diversity between cases and controls (∆π) were examined for potential candidate genes. We expect that cases would have lower π values than the SCA controls in genomic regions associated with cancer susceptibility. Therefore, we only examined windows in which this was the case. SNVs within these candidate windows were annotated using SnpEff, and resulting gene annotations were used for gene ontology (GO) enrichment analysis using g:Profiler [58] with the Ensembl 106 domestic dog annotation. The gene list was tested for significant enrichment of GO terms (FDR < 0.05) while correcting for multiple testing and considering the non-independence of GO terms. Additionally, we assessed if SNV within candidate windows were within transcription factor binding sites (TFBS) using loci from Freedman et al. [48]. The resulting candidate regions were examined further for differentiation by assessing allelic F_ST_ and differences in allele frequency. 

We also examined comparisons between cases and both control populations (SCA and SCLE). We selected the top 1% outlier loci (50 kb windows) for ∆π between cases and both control populations (SCA and SCLE), as well as the intersection of the top 1% for case/SCA control and case/both controls (resulting in 8 outlier loci). We also selected outlier peaks for each comparison for F_ST_ and ∆Tajima’s D if they were in the top 1% of normalized values. The intersection of the outlier peaks for F_ST_, ∆Tajima’s D, and ∆π were chosen as possible candidate regions. 

The resulting candidate regions were examined further for implicated SNV by assessing allelic F_ST_ and difference in allele frequency at individual SNV within each region. SNVs within these regions with an allele frequency difference greater than 0.45 (*n* = 38) were selected. The closest gene to each SNV was used as input for gene ontology (GO) enrichment analysis using g:Profiler [58]. Additionally, we assessed if these mutations were within TFBS using loci from Freedman et al. [48]. 

### 2.6. ROH Outliers

The R programming environment [57] was used for data manipulation, summary statistics, and plotting of runs of homozygosity (ROHs) using the library “detectRUNS” (https://rdrr.io/cran/detectRUNS/, accessed on 1 November 2021). Using a sliding window-based method in “detectRUNS”, ROHs were detected using a 50-SNV window to scan the genome of each individual using the LD pruned dataset. The proportion of overlapping homozygous windows to call an ROH was 0.05, with a maximum of 2 missing and one heterozygous SNV allowed within each ROH. To minimize the number of false positive ROH, the minimum number of SNVs to call an ROH was set to 2 and the minimum length of an ROH to 100 kb. A minimum density of one SNV per 50 kb and a maximum of 100 kb gap between consecutive SNV was imposed. The inbreeding coefficient, F, was estimated for each individual using FROH in the R library “detectRUNS”. 

After we detected ROH using a sliding-window method using a 50-SNV window (see above), we used “detectRuns” to calculate the proportion of times (individuals per population) each SNV falls within an ROH and plotted each proportion against the SNV position along the genome. We defined candidate loci as SNVs that were within ROHs in more than 75% of cases and less than 75% in the two control groups. 

### 2.7. MHC Genes

Several genes belonging to the major histocompatibility complex (MHC) were analyzed. Filtered SNV were selected if they were annotated with the following genes: *HLA-DRB1* (ENSCAFG00000000806), *DLA-DQA1* (ENSCAFG00000000812), *DLA_DRA* (ENSCAFG00000000803), and *DLA-DQB1* (ENSCAFG00000000814). The resulting candidate SNVs were examined further for differentiation by assessing allelic F_ST_ and ∆ allele frequencies between cases and controls. Additionally, we assessed if these mutations were within TFBSs.

## 3. Results

### 3.1. Sequencing and Genotyping

We obtained high-quality sequence reads with a per-individual average unfiltered yield of 46,553 ± 11,194 Mb (Appendix A). After removing low-quality reads and PCR duplicates, the mean sequence depth was 11.8 ± 2.3, with an average of 87% of the post-filtered genome covered by at least seven reads (Appendix A). The mean depth per population was 12.72 for cases, 11.8 for SCA controls, and 10.6 for SCLE controls (Appendix A). After filtering genotypes, we classified data into three sets consisting of variant loci (1) across all autosomes (3,945,582 SNV: Appendix A), (2) within genic regions (1,525,612 SNV), and (3) within putatively neutral regions (24,331 SNV). The transition to transversion ratio for all autosomes was 2.34, which is similar to previously reported values in wolves [48,59,60]. After LD-pruning, there were 35,406 variable positions, of which 6600 were within neutral regions. No individuals were removed due to high relatedness (parent-offspring or full siblings). 

To assess LD patterns, we estimated the physical distance at which the pair-wise genotypic association (r^2^) across all autosomal SNV decays below a threshold of 0.2. We found that populations of SCA (*n* = 23) and SCLE (*n* = 9) had moderate levels of LD (r^2^ 0.2 = ~40 kb; Appendix A). These LD levels are lower than that of other small, declining populations such as the Tasmanian Devil (Sarcophilus harrisii) ~200 kb [61] and Iberian Lynx (Lynx pardinus) ~185 kb for Andúljar [62], but r^2^ 0.2 for SCA and SCLE foxes are closer to estimates of the recently expanded Northeastern coyote population (Canis latrans) ~80 kb [63]. LD estimates may be sensitive to sample size [64] and may explain inconsistencies at longer genetic distances (275 kb). Additionally, any interspecies differences in genome structure between the reference domestic dog genome and the island fox resequencing data would result in an underestimate of the extent of LD. 

### 3.2. Population Structure and Demographic Estimates

To assess the genetic partitions based on our neutral SNV dataset, we used complementary analyses of genetic clustering patterns. The first two component axes from PCA show two distinct clusters that correspond with the sampling location (Figure 1B). On PC1, the SCLE control individuals cluster separately from SCA. Cases and SCA controls do not cluster separately based on any principal component (Appendix A). Neighbor-net analysis shows a similar pattern of strong separation between islands but no separation of cases and controls within SCA (Figure 1C). As a result of identifying the near-perfect panmixia on SCA, we did not attempt to account for background genetic differentiation between cases and SCA controls when identifying implicated loci.

We used SMC++ to estimate historical population sizes and the divergence time of these two populations (Figure 1D). The SCA and SCLE populations follow nearly identical reconstructions of Ne until about 1000 years ago, possibly representing the history of the original source population for island colonization. After this point, both populations exhibit a population contraction, followed by population recovery toward the present. The population contraction was more severe in SCLE. Each population started at ~6000 individuals and decreased to ~700 individuals on SCLE and ~1800 individuals on SCA. Although SMC++ is less accurate when estimating recent population history [50], the SCA population was at its maximum size independent of SCLE about 800 years ago and then began to decline to the present. The timing of these events and absolute population sizes, but not general conclusions, differed slightly depending on mutation rate assumptions (see Appendix A). 

### 3.3. Identification of Implicated Variants

Sliding window statistics for Weir’s F_ST_, ∆π, and Tajima’s D identified multiple regions across the genome with differences between groups (Figure 2). A window size of 100 kb (step size of 10 kb) resulted in similar outlier peaks to those at a finer scale assessment using 50 kb windows (step size of 1 kb; Appendix A). We applied XP-CLR comparisons between all cohort combinations (Appendix A). The strongest XP-CLR score in the comparison of cases and SCA controls was on chromosome 19 (Appendix A). For the XP-EHH test, there were 9481 and 8432 significant scores (*p*-value 0.001) for selection in cases and controls, respectively. The highest scores for selection in cases on chromosomes 13 and 37 and the highest scores for selection in SCA control on chromosomes 5 and 27 (Appendix A). 

Across the genome of all 32 individuals, there were 644 high-impact variants. These variants have putatively disruptive impacts on protein structure by nonsense mutation or mis-splicing. Cases exhibited 593 high-impact variants, 31 of which are not found in SCA controls. SCA controls exhibited 584 high-impact variants. Twenty-two of these were unique to SCA controls. There were 17,058 moderate impact variants, which could change protein structure as a result of an in-frame deletion or missense variant. There were 22,696 low-impact variants, which are unlikely to cause changes to protein behavior, such as synonymous variants. The remaining variants were classified as modifiers, such as variants affecting noncoding regions, where predictions are difficult or there is currently no evidence of impact. 

When assessing SNV within the top 1% of 50 kb windows for ∆π between cases and SCA controls (*n* = 21,949) after removing loci where controls had lower π than cases, 17,667 SNV remained (Appendix A). After annotation and GO analysis (number of genes = 88), one significant GO term (Neurofibromas; HP:0001067) was identified. Although no high-impact variants were found, six of these SNVs were in TFBSs. The average change in allele frequency was 0.17 with a maximum of 0.47. The mean F_ST_ estimate was 0.09, with a maximum of 0.39. With a difference in allele frequency > 0.45, only four SNVs remained (Appendix A). Of these four SNVs, two were intergenic SNVs in between ENSCAFG00000033119 (lincRNA) and zinc finger protein 536 (ZNF536). ZNF536 has been found to play a critical part in the development of lung adenocarcinoma by promoting the migration and invasion of tumor cells [65]. Another locus was an intronic SNV in DPP10 (dipeptidyl peptidase such as 10), which may play a role in colorectal cancer progression as loss of DPP10 expression in primary colorectal cancer is significantly associated with poor survival outcomes [66], and the intronic antisense lncRNA has been shown to directly affect colorectal tumor development [67].

The top 1% of outlier loci for ∆π between cases and both control populations (SCA and SCLE were found to have impacts classified as “high” (stop-gained; *n* =1) and “moderate” (missense; *n* = 72). After removing SCLE individuals, the top 1% outlier loci for ∆π between cases and SCA controls were all categorized as “moderate” impact (missense; *n* = 37; Appendix A). Of these 37 outlier loci, 8 outlier loci were found within both sets of outlier loci. One of these mutations was within TFBS on chromosome 4 (position: 6,705,047). These eight loci were assessed for allele frequencies per group (Appendix A). Several of these loci had large differences in allele frequency between SCA and SCLE, but there was no strong evidence for differences between cases and controls. 

### 3.4. Intersection of F_ST_, π, and Tajima’s D

Outlier peaks were selected if they were above the ninety-ninth percentile of normalized values from the 50 kb window scans (F_ST_: *n* = 22,005; ∆π: *n* = 9773; ∆Tajima’s D: *n* = 440; Appendix A). The intersection of these outlier peaks for F_ST_, ∆π, and ∆Tajima’s D resulted in 16 candidate regions (Figure 3). 

To identify implicated SNV, we examined allelic F_ST_ and the difference in allele frequency between cases and SCA controls within ± 150 kb of these regions (Appendix A). No fixed SNVs were identified, and the highest F_ST_ was 0.53, with an average value of 0.10. The largest difference in allele frequency was 0.59, with an average value of 0.22. SNV with the highest allele frequency difference above 0.45 (38 SNV) were annotated, resulting in proximity to 17 genes (Appendix A). Of these 38 SNVs, two SNVs were located within the intron of PTPRM (receptor-type tyrosine-protein phosphatase µ) and another two within the TFBS. PTPRM has a role in signal transduction, cell proliferation, and oncogenic transformation and is directly involved in numerous cancers, including breast [68], pancreatic [69], cervical [70], lung [71], and colorectal [72]. One SNV was identified downstream of RTN1 (reticulon 1), a biomarker for carcinomas with neuroendocrine characteristics and other cancers [73]. We found one intergenic SNV of KRT8 (Keratin 8) whose upregulation is linked to rapid tumor progression in multiple cancers, including gastric [74], lung [75], renal [76], and breast [77], and has been proposed as a pan-cancer early biomarker [78]. An intergenic SNV about 51 kb downstream of PKCδ (protein kinase c delta) was also highly differentiated between cases and controls, as well as three intronic SNVs in the TFBS. PKCδ is a tumor suppressor gene involved in the positive regulation of cell cycle progression and is also directly implicated in the progression of numerous cancers, including breast [79]. One intergenic and one upstream gene variant in the TFBS of CFAP45 (cilia and flagella associated protein 45, also termed CCDC19 and NESG1) had higher differences in allele frequency and whose downregulation is associated with lung and nasopharyngeal carcinoma [80,81]. Upstream SNV (*n* = 2) of SLAMF8 (SLAM family member 8) were highly differentiated between case and controls. This gene is responsible for lymphocyte activation. Three SNVs (2 intronic and 1 upstream) were found in FCRL6 (fc receptor such as 6), a receptor expressed by cytotoxic T and NK lymphocytes that have been implicated in immune-stimulating cancers as well as those involving PD1 (programmed cell death protein 1) [82]. This gene is involved in cell-mediated immunity by extracellular interactions with MHC II and has been identified as an immunotherapy target for multiple cancers [83]. Three SNVs were located proximal (1 downstream, 2 upstream) to DUSP23 (dual specificity phosphatase 23), which is a regulator of ERK and with implications in multiple cancers [84], had a difference in allele frequency above 0.45 between cases and controls.

These seventeen genes were used for gene ontology (GO) enrichment analysis. We found two significantly enriched GO terms, both of which were related to the regulation of superoxide anion generation (GO:0032928 and GO:0032930). The two genes that define the significant GO terms are Decapping MRNA 1A (DCP1A) and C-Reactive Protein (CRP). We found two intronic variants with allele frequency <0.45 and 2 upstream variants within the TFBS of DCP1A, which is a biomarker used for gastric [85], colorectal [86], and hepatocellular cancer progression [87]. Six variants were found downstream of the CRP gene, which is an inflammatory biomarker closely associated with prognosis in lung and gastric cancers, as well as those involving PD1 [88,89,90].

### 3.5. ROH Outliers

A total of 55,317 ROH were identified with an average of 1728.7 (±787.3) ROH per individual with a minimum of 90 and a maximum of 3283 ROH per individual (Appendix A). The average number of ROH was 1682.6 (±699.7) for cases, 1997.2 (±157.2) for SCA controls, and 1461.9 (±1231.1) for SCLE controls. There were no statistically significant differences between group means as determined by one-way ANOVA (F(2,29) = 1.1914, *p* = 0.3182; Appendix A). The average length of ROH was 0.2582 Mb for cases, 0.2527 Mb for SCA controls, and 0.2096 Mb for SCLE controls. Using FROH as calculated using “detectRUNS” in R, the inbreeding coefficient ranged from 0.04 to 0.56 (Appendix A) with an average per population of FROH = 0.32 (±0.12; case); FROH = 0.37 (±0.02, SCA control); FROH = 0.28 (±0.19, SCLE control). 

ROHs were found genome-wide (Appendix A). Figure 4 shows the proportion of times each SNV falls within an ROH for cases (Figure 4A), SCA controls (Figure 4B), and SCLE controls (Figure 4C). We identified a total of 268 ROHs, with SNV that were within ROHs in more than 70% of the individuals per population (cases: *n* = 18, SCA: *n* = 151, and SCLE: *n* = 99). Of the 18 ROHs in cases, 13 of these were not found in the control populations and were considered candidate ROHs. Two of these regions, on chromosome 4 (Figure 4D) and chromosome 21 (Figure 4E), show differentiation between case and control populations. On chromosome 4, 10 of 12 individuals (83.3%) of cases have the same region in ROH, whereas 7 of 11 SCA control individuals (63.6%) and 4 of 9 (44.4%) of SCLE controls had the same region in ROH. 

### 3.6. MHC Genes

MHC regions have notably high levels of diversity that can lead to difficulties with alignment, particularly between species [85]. The average depth for these regions was 14.71 (HLA-DRB1: 12.53, DLA-DQA1: 15.54, DLA_DRA and DLA-DQB1: 16.07), with quality (GQ) ranging from 38.08 to 50.58 (HLA-DRB1: 50.58, DLA-DQA1: 45.8, DLA_DRA and DLA-DQB1: 38.08). In human studies, diversity has been found to be so high in MHC class II, such as DLA_DRA and DLA-DQB1, that alignment is unreliable [91], which may be contributing to the lower quality score of this region in our study. Using annotations from SnpEff, 675 filtered SNVs were annotated for these selected MHC genes. A majority of the 675 SNV were classified with a putative impact of “modifier” (intergenic: *n* = 407, downstream: *n* = 95, upstream = 42, intronic: 79, 3′ UTR: 5). “moderate” impact variants included 39 missense mutations. “low” impact variants consisted of seven synonymous variants and one splice site variant. 

We examined the difference in allele frequency between cases and SCA controls at SNV within the MHC region (Appendix A). No fixed SNVs were identified, and the highest F_ST_ was 0.31, with an average value of 0.004234. The highest value of the difference in allele frequency was 0.48 (position 2208261), with an average value of 0.08027. Position 2208261, which had the highest F_ST_ and delta allele frequency, is an intergenic variant between *DLA-DQB1* and *DLA-DQA1*. Of the 675 SNVs, three SNVs were located within TFBSs (positions 2220883, 2220892, 2250952).

## 4. Discussion

This work uses genome scans based on allele frequency to identify patterns of variation enriched in foxes with ceruminous gland epithelial tumors (*n* = 12) when compared to individuals who are unaffected on the same island (*n* = 11), as well as a closely related population with mite infections that do not result in tumorigenesis (*n* = 9). The sample size does not lend itself well to traditional genome-wide association studies, and therefore an outlier approach was undertaken in the context of selection against deleterious variation as the populations rebound from severe population bottlenecks. Multiple genes implicated in cancers have been shown to be under selection in mammals [20]. Though some are the result of gene birth and death between species, this work sought to identify regions implicated in cancer susceptibility in 1:1 orthologous regions between the dog and fox genomes. Exploration of copy number influenced regions would require both higher sequencing depth (~25× average coverage) and a high-quality reference genome for the *Urocyon* grey fox genus.

### 4.1. Demographic Estimates

Our historical population size estimates for SCLE and SCA foxes began to diverge roughly 1000 years ago (Figure 1), which may represent the history of the original source population for the island colonization. Our timing estimates for changes in Ne may be affected by an imprecise mutation rate, and an underestimate of average generation time, or a lag in divergence in effective population sizes once the populations were established on each island. The Ne between the two island populations begins to differ more drastically around 470 generations ago, which is generally consistent with other evidence for the colonization history of the islands. Hofman et al. (2015) estimated the date of the earliest Urocyon island fossil to be between ~5500 to ~5700 years ago based on Accelerator Mass Spectrometry (AMS) 14C dating [26]. Interpreting this rooted phylogeny, a reanalysis of Hofman’s mitogenomes deduced the origin of the northern island foxes appears to originate from a single matrilineal founder with the southern islands with multiple, independent maternal founders either from the northern islands or mainland [92]. Other archeological records show that foxes first appeared on Santa Catalina Island sometime between 3880 and 800 years B.P., which corresponds with the length of time that the Little Harbor Site was inhabited by humans [93]. Our data show a more dramatic decline in the SCLE population from ~5000 to ~3000 years ago. Radiocarbon-dated bones suggest that island foxes were introduced to this island by Native Americans >4300 B.P. but before 3400 B.P., also corresponding to the length of time that the island is believed to have been occupied [94]. Sluggish recovery from founder effect/bottlenecks and drift as well as inbreeding could have led to decreasing Ne after divergence from other populations. There are many factors that can influence demographic estimation, particularly among island populations that interact with humans. Future explorations of demography should consider the potential effects of admixture, overall heterozygosity, and directionality of colonization (it from island to island or mainland to island). Reanalysis will be possible with the use of an interspecies reference genome, increased sampling, and higher coverage of resequencing data.

### 4.2. Multiple Candidate Genes for Cancer Susceptibility

We found no evidence for a single major effect locus for cancer susceptibility. In contrast, we identified a relatively large number of candidate loci, some showing association with cancer, which is consistent with the understanding of evolution and polygenic cancer susceptibility in humans [95]; and we found 17 genes with putative functional relationships to cancer, with an emphasis on inflammation-induced neoplasm. Our limited sample size of cases and controls reduced our ability to quantify the relative effect sizes or proportion of phenotypic variance explained by these loci, so we did not produce an estimate of heritability. 

Although we did not find any fixed differences between cases and controls, we found several candidates that have high support as outliers from multiple independent tests, each with their own unique assumptions (Appendix A). Given the potential problem of over-analyzing candidate genes [96], we discuss these loci in Appendix A and add additional comments on these candidate genes. Most of our candidate SNVs were found within intergenic and intronic regions and some within TFBSs. Many variants associated with cancer risk are located in intergenic and intronic regions with unknown functions [97]. This can make interpretation difficult but may suggest that modification of gene regulatory regions may contribute disproportionately to the modulation of risk (Appendix A).

### 4.3. Alternative/Complementary Causes of Disease

Alternatively, although outside of the scope of this study, cancer susceptibility may be due to other genetic factors than SNV. Calling structural variants was out of the scope of this study due to complications of aligning fox sequences to the domestic dog genome that may lead to erroneous inferences, as well as a lack of sufficient read depth. In the future, structural and regulatory variants should be considered when assessing the susceptibility of cancer in SCA foxes. Further, epigenetic factors, heritable changes in the regulation of gene expression that are not accompanied by changes in DNA sequence, can contribute to the pathogenesis of cancer by altering DNA accessibility. Abnormalities in methylation, histone modification, nuclear topology, and noncoding RNA have been implicated in the silencing of key tumor suppressors, regulatory, and repair genes resulting in cancer [98]. Genetic, environmental, or metabolic insults may have induced overly restrictive or overly permissive epigenetic factors that led to increased cancer risk in SCA foxes.

In addition to the inherited increased risk of cancer, as we have proposed here, differences in the tumor microenvironments within the ear canals of affected SCA foxes may be a vital aspect of tumorigenesis and progression [99]. Recent studies have emphasized that cancers are heterogeneous collections of cells that evolve in tumor microenvironments with complex ecologies, which is particularly true of cancers influenced by immune system modulation [100,101]. Mite populations, undetected toxins, and/or microbial community alterations due to mite infection may affect this tumor environment. Indeed, when SCA foxes were treated for mite infections, not only were mite loads greatly reduced, but treated foxes had reduced hyperplasia compared with untreated controls [36]. Researchers proposed that the long-term presence of mites is associated with epithelial hyperplasia, and in support of causality, removal of the parasite burden resulted in the reversal of tumor development. Further exploration of the microbiome of the island fox ear canal, tumor microenvironment, and inflammatory response could increase our understanding of the mechanisms involved in cancer development [35]. One study has found an association between mite infected versus healthy SCA foxes and the microbiome composition within ear canals; mite-infected ear canals exhibited lower microbial diversity and higher relative abundance of *S. pseudintermedius* [102]. Microbial community composition may play a role in prolonged inflammation progressing to cancer. In addition to its use as a pan-cancer biomarker and expression associated with multiple cancers [74,75,76,77,78], KRT8 maintains gut microbiota homeostasis as well as reduces colonic permeability, which is important in protecting against inflammation leading to colitis and colitis-associated tumorigenesis [103]. It is worth pursuing that a similar mechanism is present in the development of ceruminous gland carcinoma in SCA foxes. 

Alternatively, perhaps there is something unique about the SCA environment resulting in an elevated prevalence of carcinoma in SCA foxes. Little is known about the difference in prey consumption (e.g., mice, prickly pear cactus) between island foxes [104]. Habitat quality, such as habitat mineral levels and environmental pollution, may differ between islands, which may contribute to cancer susceptibility in SCA foxes. Further, Catalina Island uniquely has a substantial presence of tourists and permanent human inhabitants relative to other islands. This, as well as several non-native species such as bison and mule deer, may increase stress and subsequently affect immune responses leading to the increase in tumor prevalence in the SCA population. It is possible that a combination of genetic risk factors and environmental stressors specific to this island influence the prevalence of this condition.

### 4.4. Conservation Implications

Substantial morbidity and mortality in many wildlife species may be due to cancer [11,13]. In domestic animals, anthropogenically-induced population bottlenecks and selective breeding appear to contribute to oncogenic processes [105]. Genetic drift, as well as other anthropogenic impacts such as pollution, may also influence cancer development in wild populations [10,106]. In the case of Santa Catalina Island foxes, a human-induced viral epidemic led to a bottleneck. The reduced effective population size and increased effect of drift may have contributed to an increase in cancer prevalence. Here, we have used genomic scans to document, for the first-time, numerous regions of the SCA genome that are associated with cancer susceptibility. Our finding that cancer susceptibility appears polygenic can inform genetic monitoring and conservation of this population over time. Ultimately, the development of a susceptibility panel of genetic loci based on identified genes and genetic pathways would provide an efficient way to genetically assess an individual fox’s probability of developing cancer as a hyper-immune response to mite infection. Because this trait is likely due to the effects of many moderately deleterious mutations, the number of loci required for a susceptibility genetic marker panel would need to be greater than if large effect alleles were implicated. Little is known about how to genetically monitor quantitative maladaptive traits using cost-effective methods. However, an increased sample size combined with novel method development involving the combination of genomic scans and machine learning may reveal more about the molecular basis of this polygenic disease risk and allow for more accurate prediction of individuals with an elevated risk of cancer [107]. Long-term monitoring of susceptibility alleles, as well as overall genetic diversity, will provide a crucial data source for the long-term persistence and management of island foxes. 

Although it is well supported in applied conservation that there are often positive effects of population augmentation with divergent immigrants [108], it is often difficult to predict if the benefits of genetic rescue outweigh the potential risks of outbreeding depression or if natural purging of deleterious alleles via evolutionary rescue would have resulted in the desired outcome without the extensive resources often required of genetic rescue [6]. Recent genomic assessments of island foxes have shown that this species had likely purged its burden of strongly deleterious alleles, reducing the risk of inbreeding depression [1,2]. In fact, the morphological assessment indicates that island foxes do not exhibit many canonical signs of inbreeding depression or genetic load [1]. Cancer susceptibility in SCA foxes may be an exception to this pattern, despite the elevated genetic diversity in SCA as compared with other island fox populations. 

Evolutionary rescue (adaptation resulting from existing genetic variation) may be effective in reducing cancer susceptibility, especially since the population size has recovered from the recent bottleneck. We found no evidence of candidate loci fixed for alternative alleles between groups, suggesting that the population maintains allelic variation that could allow for an adaptive response to the disease, even if the selection is relatively weak on each locus. This implies that attempts to increase adaptive potential by translocating individuals from other islands are unlikely to have a large benefit. We suggest further field and genetic monitoring illuminate trends in cancer prevalence as well as simulations of purifying selection and drift to predict the future course of population-level susceptibility under different scenarios. 

An increasing body of knowledge is emerging with regard to conservation and management of adaptive potential [109]; yet, theory and applied studies of genetic management to confront situations where reduced population fitness is the cumulative result of many small-effect loci rather than large-effect loci, remain scarce. Due to the complexity of polygenic diseases, embracing recent advances in other fields, such as quantitative genetics, human genetics, epigenetics, and expression profiling, should result in better-informed applications of conservation genomics than have been previously possible for most wild organisms [110]. With regards to SCA foxes, further studies are needed to decipher the underpinnings of cancer risk, inbreeding, and infection/inflammation, particularly in a changing environment. Approaches that examine the genomics of tumors, combined with an understanding of the ear canal microbiome, tumor microenvironment, and immune response, are necessary. Each of these would benefit from a highly contiguous and accurately annotated reference genome for the genus *Urocyon*.

## Figures and Tables

**Figure 1 genes-13-01496-f001:**
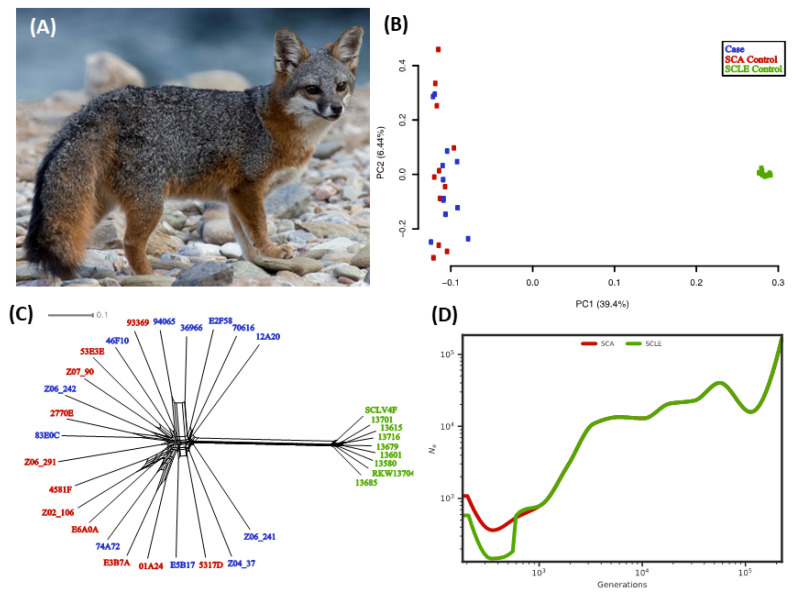
Catalina Island fox demographics. (**A**) Healthy island fox (photo credit Jaymi Heimbuch). Patterns of differentiation and divergence between populations using (**B**) PCA, and (**C**) Neighbor-net analysis. Case individuals are shown in blue, Santa Catalina Island (SCA) controls in red, and San Clemente Island (SCLE) controls in green. (**D**) Effective population size (Ne) over time calculated from single nucleotide variant (SNV) genotypes within neutral regions. A 2 year generation time and a mutation rate of 2.0 × 10^−8^/site/generation were assumed [2]. SCA individuals are shown in red and SCLE individuals in green.

**Figure 2 genes-13-01496-f002:**
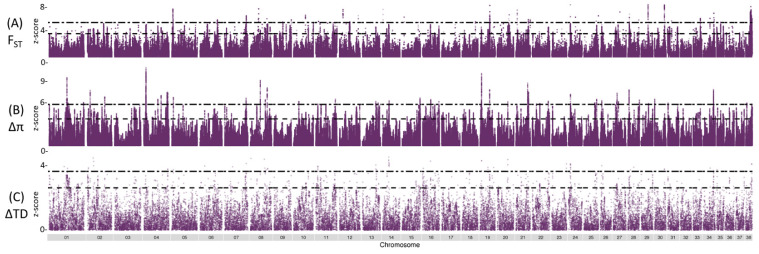
Genome-wide allele frequency SCA case (*n* = 12) and SCA control (*n* = 11) individuals from overlapping 50 Kb windows in 1 Kb steps. (**A**) Manhattan plot of z-transformed F_ST_ values. Dashed line indicated top 1% (z-score = 3.48). Double dashed line indicated top 0.1% (z-score of 5.41). (**B**) Distribution plots of ∆π values. Dashed line indicated top 1% (z-score = 3.77). Double dashed line indicated top 0.1% (z-score of 5.81) (**C**) Distribution plots of ∆Tajima’s D values. Dashed line indicated top 1% (z-score = 2.64). Double dashed line indicated top 0.1% (z-score of 3.67).

**Figure 3 genes-13-01496-f003:**
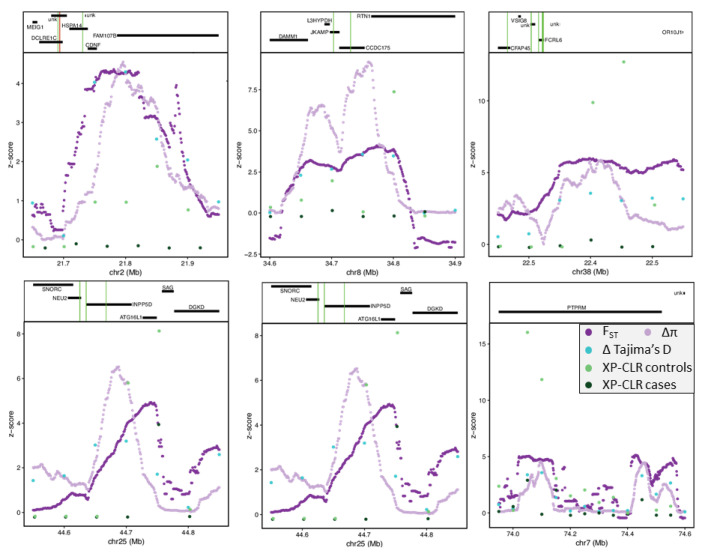
Z−transformed selection scan statistics (**bottom**) and gene annotations (**top**) plotted across the top 16 ranked candidate regions highly differentiated between case and SCA controls. F_ST_ (purple), ∆π (lavender), ∆Tajima’s D (turquoise), and XP-CLR (CO: control (light green); CA: case (dark green)). Gene annotation show genes as black horizontal bars with missense mutations as green vertical lines and stop-gained mutations as red vertical lines.

**Figure 4 genes-13-01496-f004:**
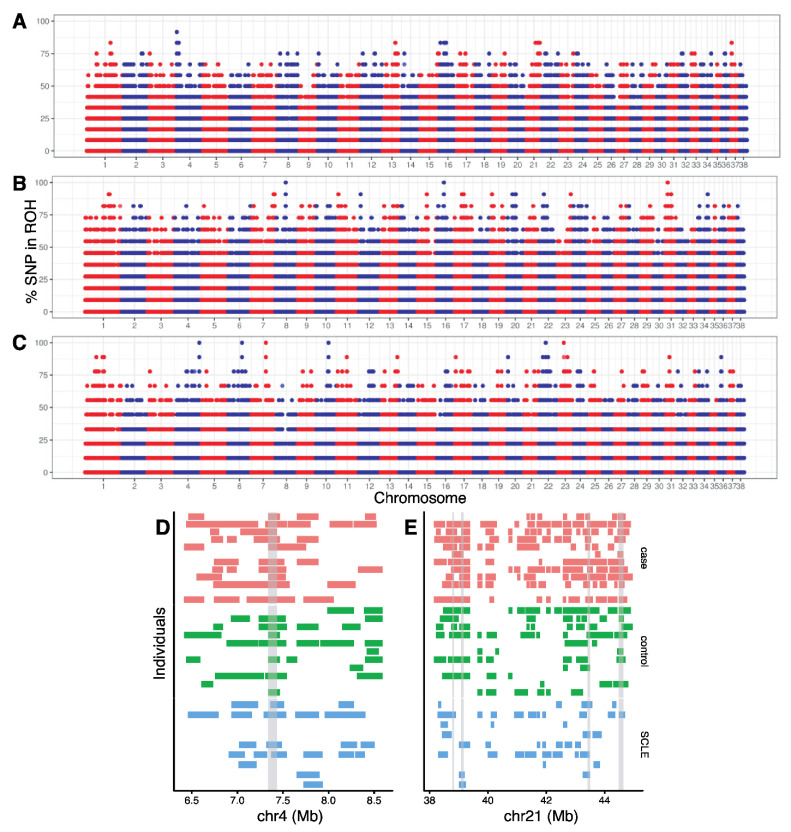
Manhattan plot of the proportion of times each SNV falls within a ROH in the (**A**) cases, (**B**) SCA controls and (**C**) SCLE controls. Chromosomes 1–38 are arranged left to right, with alternating red and blue representing different chromosomes. (**D**) ROH detected in each individual on a part of chromosome 4 and (**E**) chromosome 21. Gray bars represent regions where 75% of cases (shown in salmon) have overlapping ROH and less than 75% of SCA controls (shown in green) and SCLE controls (shown in light blue). See Appendix A for genome-wide ROH for each individual.

## Data Availability

Raw sequencing data are available from the NCBI Sequence Read Archive under the Bioproject PRJNA870583.

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
