# Peer review of "Genomic Assessment of Cancer Susceptibility in the Threatened Catalina Island Fox (*Urocyon littoralis catalinae*)"

_genes, 2022, doi:10.3390/genes13081496_

Round 1
Reviewer 1 Report
Overall, I thought this was a clear and novel comparison of genomes between Catalina Island foxes affected versus not affected with tumors (and with San Clemente Island foxes, for which the population was tumor-free). Although no single mutation was strongly implicated, the comparison turned up a number (17) of potential candidate genes potentially conferring vulnerability to tumors and suggesting that susceptibility was likely polygenic. As authors point out, the sample size was underpowered for a GWAS, limiting their ability to quantify heritability. I appreciated that authors did not overinterpret findings and presented a balanced treatment of alternative causes in the Discussion. I would have liked to see a bit of discussion about how the age of onset of the disease might affect both its influence on population growth and, therefore, strength of selection. I appreciated the section on Conservation Implications, including general issues of genetic monitoring when many loci contribute to disease susceptibility. The final recommendations with respect to evolutionary rescue (counting on natural selection on standing genetic variation to reduce frequency of deleterious alleles) verses risks of outbreeding depression if individuals from other islands were introduced seemed well-thought out, logical, and consistent with findings.
I was unclear as to the relevance of the historical demographic analysis. While I found it interesting for completely unrelated reasons, I did not understand how it contributed to the objectives of the current paper. I have no objection to its inclusion, but if it is to be retained, some corrections would seem essential. In particular, the timescale indicated on the figures (1D, Supp Info) are an order of magnitude different from those in the text. For example, in figure 1D, the traces of the two island populations split ~1,000 (10^3) generations ago, which, in the text, is reported as ~10,000 years ago. This is a very substantial and meaningful difference. Consequently, if the figure is correct, all of the text on this topic in the Results and Discussion should be completely reworked. It would actually be much more interesting if the figure is correct because it suggests a much more recent shared ancestry between two of the southern islands than is typically assumed. If the text is correct, however, the figures need to be corrected. In either case, I would also suggest a more cautious interpretation of traces with respect to population divergence. I appreciated the attempt to accommodate uncertainty in mutation rate, but other factors can distort historical demographic reconstruction, including admixture (e.g., via human translocations), low coverage (as applied to the current data set), runs of homozygosity (again, applicable), and simply the possibility of two populations undergoing parallel demographic trajectories despite being on different islands. As authors are aware, there are other more direct ways to assess phylogenetic divergence. I am not suggesting the paper needs any additional analyses, simply that interpretations be caveated.
Otherwise, I have made several comments, mostly minor, corresponding to specific statements:
Lines 57-59: an illustrative example would be helpful, i.e., showing how a disease that typically affects older individuals have significant effects on population reproduction and growth.
Line 77: please add the scientific name of mainland gray fox
Line 78: Please change the wording so that it does not imply a stepping-stone model of island colonization as we do not know the true history and other models are possible. Although I agree that (a) the island fox occurred earlier on the northern than southern Channel Islands (and your time frame is about right) and (b) that genomic data appeast tp suggest that all island foxes share more recent ancestry with one another than with mainland gray foxes (e.g., Robinson et al. 2018), we do not know that the island fox of the southern islands derived from those of the northern islands. In fact, the mitochondrial data suggest otherwise (see discussion and reanalysis of Hofman et al. mitogenomes by Goddard et al. 2015). For example, it is possible that a now-extinct population, even a semi-domesticated or human-commensal population, existed on the mainland from which both northern and southern populations were drawn at different times and by different means.
Line 89: Please soften your assertion about the role of balancing selection affecting MHC in San Nicolas Island foxes add the reference by Hedrick (2005, Heredity, 93:237-238) that exposed some significant problems with the conclusions of Aguilar et al. (2004).
Line 140: not clear what you mean by phred score > 20 here in the context of haplotype caller. Is this meant to be mapping quality or base quality or other? Please specify.
Line 142: confusing to say that final dataset included sample fitting a minimum depth and then indicate that that composed 32 foxes as that seems to be the entire dataset. Maybe clearer to delete this statement entirely and simply report the depths in the Results.
Lines 142-147: suggest deleting this paragraph and integrating the relevant parts into the “Library construction and sequencing” section above when you first introduce the samples you used. Here, it breaks interrupts your bioinformatic workflow.
Lines 202-204: Please clarify what you mean by “lower values.” Do you mean that cases are expected to have lower nucleotide diversity than controls?
Line 216: You might hold off mentioning that there were 8 outliers until the results. If not, then I suggest you add the same information to the previous paragraph for consistency.
Line 217: I understand the value of Tajima’s D if you are looking for a signal of positive selection and if you were looking throughout the entire population (not cases and controls independently), but I am not clear on what you are predicting and why for Tajima’s D calculated separately for cases and controls. I suspect you have a good reason, but could you clarify the logic in the text?
Lines 218-219: looks like you left out a word here. As written, it is an incomplete sentence.
Lines 255-259: Please round depths to 1 decimal place and add SD to provide a sense of variance.
Lines 288-290: I do not understand this statement. The traces are nearly identical until <1,000 generations ago (i.e., 10^3 on Fig. 1D). Is it more likely that this indicates a more recent shared ancestry than previously thought? Why not? The Gabrielino Indians interacted with these foxes on the three southern Channel Islands and could have moved them among islands around that time (see various writings of Paul Collins).
Lines 292-293: Again, the timing on the graph does not agree with that in the text here. Off by approximately an order of magnitude.
Line 296-297: I do not see a “peak” at 800 years, nor do I see one anywhere on the trace after which the decline extends to the present. Please clarify.
Line 347: Abbreviate as TBFS for consistency.
Figure 3. The font is way too small to read on these graphs.
Line 366: “Of these 38 SNVs,…” Not clear what you are referring to. You have just mentioned above 16 candidate regions, which include 17 genes. Where did the 38 come from?
Figure 4. Minor point: Methods indicate 70% as the threshold, but the figure caption and Results text refer to 75%. More importantly, unless I am missing something, it seems the analysis seems underpowered to be very meaningful. The SNVs shown that occur in >75% of ROH in cases but <75% of ROH in controls (particularly on SCA) do not differ much at all in frequency between cases and controls, suggesting that haplotypes in these regions have low diversity island-wide, resulting in high homozygosity regardless of cancer status.
Lines 463-464: see comments above regarding lines 288-290
Line 469: rather than “4,700 years,” I believe this should be 470 generations or 940 years.
Lines 470-472: Re: “Hofman et al. (2015) estimated that southern island lineages diverged from each other ~5500 to ~5700 years ago based on Bayesian phylogeny of mitogenomes and Accelerator Mass Spectrometry (AMS)14C dating[87]” This is not quite accurate. Hofman et al. estimated the date of the earliest fossil on one of those islands to be between 5500-5700 years but then used that as a basis to calibrate their mitochondrial tree. There was no independent mtDNA evidence of that divergence timeframe. In fact, Goddard et al. (2015; Plos one, 10[8], e0136329) pointed this out and presented revised dates based on a reanalysis of Hofman et al.’s data and assuming mutation rates derived from other canids. Their conclusion was that all mitogenomes of the northern islands could be potentially traced a single matrilineal founder, but not the southern islands, which represented at least two if not three (one for each island) distinct maternal founders.
Lines 475-476: timeframe again disagrees with figure.
Lines 494-494: incomplete sentence
Author Response
See attached response to reviewers

Reviewer 2 Report
In their paper, Hendricks and coauthors address a topical conservation question, namely how cancer affects genetically depleted small population that have experienced genetic drift and exhibit a small population size. For this purpose, they select an insular population of a small canid of conservation and evolutionary interest, the near threaten Catalina Island Fox (Urocyon littoralis catalinae). The study is well-written and organized, well-discussed and well-illustrated; the laboratory and computational analyses are novel, the results very robust; the cited literature comprehensive and pertinent.
I have just minor observations to make, most of them dealing with oversights spanning from the correct usage of acronyms and scientific names to the consistency with notations used (i.e. FST) and some typos.
For reason of practicality, I have added a number of edits in the attached pdf; the authors will find a suggestion for change or comment by placing the cursor on the edited text.
Importantly, the newly produced genomes should be submitted to GenBank or any other public repository for other researchers to access these resources upon their needs.
Note that in the Supplementary Files you have used "et al." both in italics and not.

Author Response
We thank the reviewer for the comments on the manuscript. We have uploaded the responses as comments on the PDF marked by the reviewer. We also want to convey that we wish all reviewers would mark up documents in this fashion for review, as it makes the editing process immensely better.
Regarding the submission of sequencing data to a repository, we listed a bioproject accession placeholder at the bottom of the manuscript. This will be released from embargo when the manuscript is published
